# Changes in Homogalacturonan Metabolism in Banana Peel during Fruit Development and Ripening

**DOI:** 10.3390/ijms23010243

**Published:** 2021-12-27

**Authors:** Tong Ning, Chengjie Chen, Ganjun Yi, Houbin Chen, Yudi Liu, Yanjie Fan, Jing Liu, Shule Chen, Sixuan Wei, Zexuan Li, Yehuan Tan, Zhenting He, Chunxiang Xu

**Affiliations:** 1Department of Horticulture, College of Horticulture, South China Agricultural University, Guangzhou 510642, China; ningtong@stu.scau.edu.cn (T.N.); ccj@scau.edu.cn (C.C.); hbchen@scau.edu.cn (H.C.); liuyudi18@mails.ucas.edu.cn (Y.L.); fanyj@webmail.hzau.edu.cn (Y.F.); liujing@stu.scau.edu.cn (J.L.); chsl1113@stu.scau.edu.cn (S.C.); sixuanwei@stu.scau.edu.cn (S.W.); yy2018lzx@stu.scau.edu.cn (Z.L.); yehuantan@stu.scau.edu.cn (Y.T.); hezhenting@stu.scau.edu.cn (Z.H.); 2Institute of Fruit Tree Research, Guangdong Academy of Agricultural Sciences, Guangzhou 510640, China; yiganjun@vip.163.com; 3Guangdong Labortatory for Lingnan Modern Agriculture, Maoming Branch, Maoming 525000, China

**Keywords:** banana (*Musa* spp. AAA), fruit development and ripening, fruit peel, homogalacturonan metabolism, immunofluorescence labeling

## Abstract

Though numerous studies have focused on the cell wall disassembly of bananas during the ripening process, the modification of homogalacturonan (HG) during fruit development remains exclusive. To better understand the role of HGs in controlling banana fruit growth and ripening, RNA-Seq, qPCR, immunofluorescence labeling, and biochemical methods were employed to reveal their dynamic changes in banana peels during these processes. Most HG-modifying genes in banana peels showed a decline in expression during fruit development. Four polygalacturonase and three pectin acetylesterases showing higher expression levels at later developmental stages than earlier ones might be related to fruit expansion. Six out of the 10 top genes in the Core Enrichment Gene Set were HG degradation genes, and all were upregulated after softening, paralleled to the significant increase in HG degradation enzyme activities, decline in peel firmness, and the epitope levels of 2F4, CCRC-M38, JIM7, and LM18 antibodies. Most differentially expressed alpha-1,4-galacturonosyltransferases were upregulated by ethylene treatment, suggesting active HG biosynthesis during the fruit softening process. The epitope level of the CCRC-M38 antibody was positively correlated to the firmness of banana peel during fruit development and ripening. These results have provided new insights into the role of cell wall HGs in fruit development and ripening.

## 1. Introduction

Bananas (*Musa* spp.) are one of the most important tropical fruits and food crops in the world. It is commonly harvested at the mature-green stage when the angularity on the fruit surface disappears. Banana is a typical climacteric fruit, and exogenous ethylene is usually used to artificially initiate fruit ripening prior to commercialization. During the ripening process, there is a burst of ethylene production coordinated with rapid softening-associated processes, including peel de-greening, peel, and pulp softening [1]. In the meantime, the tolerance/resistance of fruits to both mechanical damage and microbial infection will decrease [2], which usually causes enormous economic losses. In addition, the fruit properties formed during the development stages are also closely related to its stress tolerance/resistance and mechanical characteristics after harvest [3]. Therefore, a better understanding of the process of fruit development and ripening may help to improve the nutritional and sensorial quality of banana fruit and reduce post-harvest fruit losses.

Plant cell walls play a fundamental role in fruit development and ripening. It is a rigid structure contributing to fruit firmness, and its modification during fruit development and ripening affects fruit texture [4,5]. Fruit softening is generally attributed to the disassembly of the cellulose and hemicellulose network through the depolymerization of pectin and hemicellulose [6]. Many genes and enzymes/proteins related to the biosynthesis [7,8] and degradation of cell wall components [9,10,11,12,13,14,15] are reported to be involved in fruit development and ripening, besides non-enzymatic proteins (e.g., expansin) [16,17].

Pectin is one of the major components of the primary cellular walls and middle lamella in plant tissues, making up 30–35% of primary cell walls in dicotyledonous species and non-graminaceous monocots, 2–10% of grass and other commelinoid primary walls, and up to 5% of walls in woody tissue [18]. It is one type of highly complex polysaccharides and could be classified into four major types, namely homogalacturonan (HG), rhamnogalacturonan I, rhamnogalacturonan II, and xylogalacturonan [19,20]. HG is the most abundant pectic domain, comprising 55–70% of pectin. It is a linear homopolymer of α-1,4-linked galacturonic acid (GalA) that is partially methyl-esterified at C-6 and acetylated at O-2/O-3 [19]. HG is synthesized in Golgi and produced into the cell wall in an extremely methyl-esterified arrangement. Alpha-1,4-galacturonosyltransferase (GAUT; EC 2.4.1.43) catalyzes the transfer of GalA from uridine-diphosphate-GalA (UDP-GalA) onto HG acceptors [21,22]. A more recent study revealed that the GAUT1: GAUT7 complex is the catalytic core of an HG: GalAT complex [23]. The HG degradation enzymes mainly included polygalacturonase (PG; EC 3.2.1.15), pectin methylesterase (PME; EC 3.1.1.11), pectate lyase (PL; EC 4.2. 2.2), and pectin acetylesterases (PAE; EC 3.1.1.6) [24,25]. Among them, PMEs are the key enzyme controlling both the assembly and disassembly of the pectic network. The predominant role of PMEs is to remove methyl groups from contiguous GalA residues on an HG backbone, which results in relatively long stretches or blocks of de-esterified residues. PME action is required before PGs or PLs can act on methyl-esterified HG chains [19,20].

HG degradation is believed to play an important role in fruit ripening, which leads to the disassembly of cellulose and hemicellulose network and a decrease in fruit firmness. For example, transgenic fruits with lower expression levels of HG degradation genes (e.g., PLs and PGs) were significantly firmer than control ones and displayed a reduction in solubilization and depolymerization of polyuronides [26]. However, the information on the involvement of HG biosynthesis-related genes is limited.

In the case of bananas, multiple studies have been performed to investigate the mechanism involved in banana fruit ripening at physiological, biochemical, and molecular levels during the past decades. Most studies have been focused on gene expression [27,28,29,30] and activities of enzymes associated with cell wall modification [27,29,31,32,33,34], as well as the changes in different types of pectin polysaccharides [30,35]. Similar to other fruits [36,37], high throughput techniques, such as proteomics [9,10,11,13], transcriptomic [15,38,39], metabolomics [40], microRNA [41], or the combination of different omics [12,14], have also been employed to provide insights into genes, proteins, and metabolites involved in banana ripening based on the release of banana genomics database [38]. A large number of differentially regulated genes/proteins were identified during banana fruit ripening, and many of them are involved in cell wall metabolism [9,10,11,12,13,14,15,38,39].

Though cell wall disassembly regard to banana softening has been extensively studied, little is known about cell wall construction during early fruit development. This study was to analyze the changes in HG-modifying genes in the peel of a Cavendish dwarf banana cultivar ‘Huanong Aijiao 1’ (*Musa* spp. AAA) at different developmental stages and the ripening process based on the whole-transcriptome gene expression analysis. In addition, an immunofluorescence labeling technique was employed to reveal the changes in the spatial and temporal distribution of different HG components in banana peels during developmental and ripening stages, besides the changes in pectin content and activities of key pectin degradation enzymes. The results will provide new insights into the role of cell wall HGs in banana fruit development and ripening.

## 2. Results

### 2.1. Morphological and Firmness Analyses

The study was performed on fruits of 0 d (just emerging from the bunch), 35 d, 60 d, green matured (GM, harvested fruits, 85 d), and yellow matured (YM, 6 d after ethylene treatment) fruits. The change in appearance is shown in Figure 1A. The fruit at 0 d was cuboid-like and yellow-greenish. There was no pulp in the fruit chamber/caving. Thirty-five days later, the fruit became curved and white pulp could be observed in the pentagon cross-section. Fruits at 60 d were less curved than those at 35 d. At harvest, the peel looked green-whitish, and the ridge on the fruit peel disappeared gradually, and the caving was filled with whitish pulp. After softening, the peel became yellow and the pulp yellow-whitish. The fruit size (indicated by fruit fresh weight, length, and perimeter) increased significantly with the development of fruits (Figure 1B).

### 2.2. Systemic Analysis of the Developmental Changes in Gene Expression Levels with RNA-Seq

To get an overview of molecular changes underlying banana fruit development, we have performed transcriptome profiling at its different developmental stages, as well as ripening. The numbers of differentially expressed genes (DEGs) between every two stages were shown in Figure 2A (0 d vs. 35 d, 0 d vs. 60 d, 0 d vs. GM, 0 d vs. YM, 35 d vs. 60 d, 35 d vs. GM, 35 d vs. YM, 60 d vs. GM, 60 d vs. YM, and GM vs. YM). Expression patterns of most genes remain unchanged between every two successive stages before fruit ripening. Only 1884 (593 upregulated and 1294 downregulated), 469 (159 upregulated and 310 downregulated), and 2505 DEGs (777 upregulated and 1728 downregulated) were identified in comparison between 0 d and 35 d, 35 d and 60 d, 60 d and GM, respectively. YM is the stage most distinguished from the others, with more than 10,000 DEGs when compared to any other stages, including the comparison between GM and YM. Gene Set Enrichment Analysis reveals significant enrichment of several items related to the cell wall, such as “GO:0042545, Cell Wall Modification” (Figure 2B). During a manual inspection, we found a series of HG-modifying genes showing the most considerable changes (i.e., the largest fold change values) (Figure 2C), such as PG (Ma09_g15970, Ma02g16140 and Ma04_g02960) and pectinesterase (Ma07_g28300).

### 2.3. The Developmental Changes in the Expression Levels of HG-Modifying Genes

In total, there are 61 MaGAUTs, 64 MaPGs, 3 MaPGIs, 27 MaPMEs, 41 MaporPMEs, 47 MaPMEIs, 12 MaPAEs, and 18 MaPLs in banana Genome A.

GAUT is the key gene family associated with the biosynthesis of HG. Nowadays, GAUT1, GAUT7, GAUT13/14 were proved to be involved in the biosynthesis of pectin HG. As shown in Figure 3, in the present study, one MaGAUT gene from each subfamily described above (Ma04_g39140, Ma11_g23340, and Ma07_g18890) was differentially expressed during fruit development, and only the MaGAUT13/14 gene showed an increased expression level. On the other hand, three MaGAUT1s, five MaGAUT7s, and two MaGAUT13/14s were regulated by ethylene treatment, and seven out of them showed significantly higher expression levels in the YM fruits when compared with earlier developmental stage(s), suggesting active HG biosynthesis during the fruit ripening stage.

In total, 23 MaPGs showed differentially expressed levels during banana fruit development and 36 after ethylene treatment. Only four MaPGs (Ma09_g22850, Ma11_g02770, Ma04_g08510, and Ma07_g16630) showed a higher expression level during the fruit developmental stages, while it was 16 for the ripening stage. The expression of two MaPGIs decreased with the development and was inhibited by ethylene treatment (Figure 3).

All five MaPMEs differentially expressed during fruit development showed a decline in expression level. On the contrary, all four MaPMEs differentially expressed during fruit ripening were upregulated by ethylene treatment. Similar trends were observed with the expression of MaproPMEs, MaPLs, and MaPMEIs except for Ma08_g12230 (proPME) and Ma08_g12230 (PMEI) (Figure 3).

Nine MaPAEs were differentially expressed during banana fruit development and ripening. Three of them (Ma07_g07240, Ma07_g07250, and Ma07_g07270) showed an increased expression level with the development of banana fruits, but a decreased one after ripening. On the contrary, the expression of Ma04_g18920 decreased with fruit development but increased after softening. Another three MaPAEs (Ma04_g28350, Ma05_g20750, and Ma11_g03750) were also induced by ethylene treatment, but the other two MaPAEs (Ma02_g20280 and Ma04_g18910) were downregulated during banana fruit development (Figure 3).

To validate the results of the RNA-Seq analysis, the expression of 25 representative DEGs (one–six ones from each gene family) were analyzed by qPCR. The result indicated that it was consistent with the result from RNA-Seq (Appendix A).

### 2.4. Enzyme Activities and Fruit Properties

As shown in Table 1, the PME activity in the fruit peel at 0 d was 15.59 U. It became lower in the peel at later developmental stages but increased significantly to 21.10 U after softening. The PL activity kept stable during the early development stages but started to increase at harvest and peaked after ethylene treatment. Differently, there were no significant differences in PG activity in the banana peel till harvest, but YM fruits showed 6.60–13.90 times higher PG activity than fruits at earlier developmental stages.

The pectin content in the peel of banana fruit at 0 d was 62.70 mg/g (dry weight). It decreased with the fruit development and reached the lowest level at harvest, which was significantly lower than that at 0 d. After ripening, it increased to 57.83 mg/g, which was at the same level as that at 35 d. The degree of pectin methylesterification (DM) of fruit peel at 0 d was the highest (59.07%) while the YM fruit showed the lowest one (47.60%), but no significant difference was observed with DM during fruit development and after ripening (Table 1).

The changes in the firmness of fruit peel during development and ripening stages were also investigated. The rupture force of fruit peel at 0 d was 10.90 *N*. It showed an increasing trend with the development of fruits and peaked at 60 d, followed by a decline thereafter. The peel firmness of GM fruits was at the same level as that of fruits at 0 d. However, the peel firmness of YM fruits was only 4.93 *N*, significantly lower than that of fruit peel at all the other stages.

### 2.5. Histological Analysis

Ruthenium red (RR) was used to reveal the spatial and temporal changes of pectin in banana peels during fruit development after treatment of the section with NaOH. As shown in Figure 4A,B, the staining of latex cells was the strongest, followed by the vascular tissue and the epidermic cells. With the development of the fruits, the cells became larger, and the vascular tissue became more developed with thickened cell walls. The color of the epidermic cells and plasma membrane became pink instead of purple-redish as before. The color of the vascular tissue in the peel at 0 d was also purple-redish. It became lighter with the development of fruits and deeper again after ripening (Figure 4C–J). The changes of the demethylesterified pectin (stained with RR, without NaOH treatment) in fruit peel during fruit development and ripening were similar to that of the pectin. When compared to the section stained by RR after NaOH treatment, fewer epidermic cells and plasma membrane were stained only with RR (Figure 4K–O), but this was not the case for the vascular tissue (data not shown).

### 2.6. Changes in the Spatial and Temporal Distribution of HG Components in the Peel

#### 2.6.1. PMEs and the Epitope of 2F4

In the fruits at 0 d, the signal of banana PME antibody was nearly undetectable, with a relatively stronger one in the epidermic cells, cortex cells adjacent to it, and latex cells (Figure 5A,B and Appendix A). No significant differences were observed with the signal strength and labeling pattern at the later developmental stages and after ripening (Figure 5C–H and Appendix A). 2F4 antibody recognizes unesterified, or calcium ion cross-linked HG. The antibody signal in the peel at 0 d was not strong neither, and its labeling pattern was similar to that of the PME antibody (Figure 5I,J and Appendix A). The epitope level in the peel at 35 d increased 40%. A relatively stronger signal was present in the epidemic cells, cortex cells adjacent to it, as well as the xylem vessels (Figure 5K,L). The antibody intensity peaked in the fruit peel at 60 d (Figure 5M,N and Appendix A) followed by a decline thereafter (Figure 5O,P and Appendix A). The antibody intensity decreased to the level of the fruits at 0 d (Appendix A).

#### 2.6.2. The Epitopes of JIM5 and JIM7 Antibodies

The JIM5 recognizes partially methyl-esterified HG (up to 40%), and this epitope was mainly present in the xylem vessel, the vascular bundles, and the epidermic cells of the peel at 0 d (Figure 6A,B and Appendix A). The antibody intensity increased slightly with the development of fruits (Figure 6C–H and Appendix A).

The JIM7 recognizing epitope is HG with a methylesterification degree of 40–80%. The signal of this antibody in the transverse section of banana peel at 0 d was also very weak, with a relatively higher level in the xylem vessel and the epidermic cells (Figure 6I,J and Appendix A). The signal became stronger with the development of fruits and peaked at harvest, especially in the cortex cells adjacent to the epidermic cell and vascular bundles. The antibody intensity (21.36) was 1.74–2.76 times higher than that of fruits at earlier developmental stages (Figure 6K–P and Appendix A). However, the antibody intensity decreased significantly after ripening (Appendix A).

#### 2.6.3. The Epitopes of CCRC-M38 and LM18 Antibodies

LM18 antibody recognizes lowly methyl-esterified HG, and its labeling pattern in the peel at 0 d was similar to that of JIM5, but with a relatively higher epitope level (Figure 7A,B and Appendix A). The LM18 epitope level increased significantly in the peel at 35 d, except the xylem vessel (Figure 7C,D and Appendix A). A stronger signal could be observed with further development of fruit, especially in the tri-cellular junction (Figure 7E,F and Appendix A). But there was a decline in signal strength after ethylene treatment, which was still higher than that in the peel at 0 d (Figure 7G,H and Appendix A). The CCRC-M38 signal in banana peel at 0 d was much stronger when compared to that of the other antibodies. It was distributed in the transverse section of banana peel at 0 d, with a relatively weaker signal in the vascular bundle sheath (Figure 7I,J and Appendix A). The CCRM-M38 signal in the peel at 35 d became much stronger (the intensity increased from 22.60 to 32.48), especially in the vascular bundle sheath (Figure 7K,L and Appendix A). No significant differences in antibody intensity were observed with further development of the fruits (Figure 7M,N and Appendix A). However, the intensity in the YM fruits decreased to 13.85 (Figure 7O,P and Appendix A).

No detectable signal was observed with CCRC-M34, CCRC-M130, and LM20 antibodies in banana peel (data not shown).

#### 2.6.4. The Calculation of Correlation Coefficient

Because the epitope levels of 2F4, CCRC-M38, JIM7, and LM18 antibodies showed similar dynamic changes to peel firmness during banana fruit development and ripening. The correlation coefficient between them was analyzed. The coefficient between peel firmness and CCRC-M38 intensity was 0.9877 with *p* value of 0.012 during fruit development, 0.9611 with *p* value of 0.009 during fruit development and ripening stage. There was also a positive correlation between peel firmness and 2F4 intensity during fruit development (coefficient 0.9369, *p* value 0.063).

## 3. Discussions

Previous studies indicated that there were great differences in pectin content and the expression of HG-modifying genes between the peel and pulp during banana ripening [12,14,42,43]. Thus, the discussion here will focus on fruit peel.

### 3.1. Changes in HG Metabolism during the Development of Banana Fruits

Fruit growth involves cell divisions and cell expansion resulting from a dynamic interplay between cell turgor pressure, cell wall biosynthesis, and degradation, as well as cell wall remodeling. It has been proposed that the cell wall component of fruit exocarp/peel controlled the expansion of the developing fruits and contributed to the final size and shape of the fruit [44,45]. However, studies related to HG metabolism in banana peels at different developmental stages are not available.

In the present study, the expression levels of all differentially expressed *PGIs*, *PMEs*, pro*PMEs*, and *PLs*, as well as most *PGs*, *PMEIs*, and *PAEs* in banana peels decreased during the fruit development. However, both biochemical and histological analyses revealed that the pectin content in banana peels decreased with fruit development, and there was a significant difference between banana peels at harvest and 0 d. This might be due to the down-regulation of GAUTs (Ma10_g04860, *MaGAUT7*; Ma11_g23340, *MaGAUT7*; Ma07_g18890, and *MaGAUT1*) in the meanwhile. On the other hand, four PGs (Ma09_g22850, Ma11_g02770, Ma04_g08510, and Ma07_g16630) and three PAEs (Ma07_g07270, Ma07_g07240, and Ma07_g07250) showed higher expression level at later development stages than earlier ones. They might be related to the expansion of fruit and provide the flexible, extensible mechanical support that resists the strains resulting from the expansion of the underlying cortex tissue [46,47]. The differential expression of PGs and PAEs or other pectin-related genes was also observed during peach (*Prunus persica*) fruit [37] and banana pulp development [30]. Concomitant expression of genes involved in HG biosynthesis and degradation, as well as differential expression patterns among the members in each HG-modifying gene family, are required to achieve a proper cell wall organization and organ development [46,48].

The firmness of fruit peels increased significantly with the development of fruits and peaked at 60 d, followed by a decline at harvest. This decline in firmness of banana peel was suggested to be due to the significant increase in PL activity. Interestingly, the IF data revealed that the epitope levels of JIM7 and LM18 antibodies increased with the fruit development and peaked at harvest, while those of 2F4 and CCRC-M38 antibodies peaked at 60 d. In addition, there was a significantly positive correlation between peel firmness and the CCRC-M38/2F4 intensity during fruit development, suggesting that the content of fully de-esterified HG and unesterified/calcium ion cross-linked HG recognized by CCRC-M38/2F4 antibody, instead of pectin content and DM, was significantly correlated with the firmness of banana peel during fruit development. The increase in the antigen level recognized by the JIM7 antibody was also observed during the early development of tomato pericarp [49].

### 3.2. Changes in HG Metabolism in Banana Peel during the Fruit Ripening Process

Banana is a climacteric fruit. It is well known dramatic changes at physiological, biochemical, and molecular levels, including peel de-greening, fruit softening, and flavor change during the ripening stage, concomitantly with a burst of ethylene production and respiration, took place during the ripening stage of this type of fruits. Among them, the most apparent change is the reduction in fruit firmness. Cell wall disassembly is considered to be the major determinant of fruit softening during the ripening process in fleshy fruits [50,51], which results from a decrease in cell turgor pressure as well as cell wall polysaccharide remodeling and metabolism [52].

Over the last decades, the dynamic expression profiles of cell wall-modifying genes and corresponding changes in enzyme activities and cell wall composition revealed that pectin modification is primarily responsible for the progressive loss of firmness in ripening and withering fruits [51,53,54]. In the case of bananas, the involvement of PMEs, PGs or/and PLs in the disassembly of HGs during banana fruit ripening has been extensively investigated. For example, Mbéguié-A-Mbéguié et al. [28] investigated the expression of nine pectolytic genes in the median area of peel tissue during post-harvest ripening of Cavendish banana fruit. They found that the expression of seven genes (PME3, PG1/2/3/4, and PL1/2) increased greatly at the late-ripening stages, but PME1 was on the contrary. Recently, some studies have been undertaken to elucidate the cell wall changes in the peel during banana fruit ripening and softening via OMICs. At the protein level, nearly all reported HG-modifying proteins showed higher expression levels in the soft banana when compared to GM banana fruits. For example, Du et al. [10] found that PL1 (gi6606532), PL2 (gi6606534) proteins were upregulated 1 d after ethylene treatment. Similarly, probable PL5 (Ma06_p30000.1), PAE 12(Ma05_p20750.2), probable PMEI12 (Ma03_p05670.1), 2 probable PGs proteins (Ma11_p02770.1, Ma04_p08510.1), and putative PG At1g48100 (Ma02_p16140.1) in the fruit peel of *Musa* AAA cv. Brazilian were found to be upregulated after ripening via ethylene treatment [13]. However, this was not the case at the transcript level. Some HG-modifying genes were upregulated, while the others were on the opposite after ethylene treatment [14,15,38]. Similar results were observed in the present study. The differentiation in the expression within HG modification gene families during the ripening stage was also reported in some other fruits [55,56,57,58]. However, in the present study, the top 10 genes in the Core Enrichment Gene Set included three PGs, one PME, and two proPMEs, and all of them were upregulated by ethylene treatment. These results further confirmed that the HG-modifying genes were the major determinant for the disassembly of pectin polysaccharides in banana peels during the fruit ripening stage.

Nowadays, it is proved that GAUT1, 7, 13, and 14 are involved in the biosynthesis of HG [23,59]. In the present study, 5 *MaGAUT7s*, three *MaGAUT1s*, and two *MaGAUT13/14s* were differentially expressed during banana fruit ripening. Seven out of these 10 *MaGAUTs* were upregulated by ethylene treatment, suggesting there was still active biosynthesis of pectin HG during this process, which was in accordance with the observation in the ripening process of apple (*Malus domestica*) [55].

Consistent with the gene expression pattern, the activity of PG and PL in the peel of YM fruits was 2.25 and 6.60 times higher than that of GM fruits, respectively. Additionally, the PME activity also increased significantly from 13.09 to 21.10 in the present study. As a result, the fruit firmness decreased significantly after softening. Previous studies also found the activity of PG and/or PL in banana peels increased [29,32,33], but the PME activity kept stable [29] during post-harvest ripening. The essential or even central role of plant PMEs, PGs, and PLs in the fruit softening process have been confirmed in several plants [4,60,61,62,63,64].

Besides, at the molecular level, fruit cell walls also undergo extensive changes in structure and composition during fruit ripening, which results in the reduction in their mechanical strength. In the case of bananas, Duan et al. [35] suggested that the rapid firmness decrease in banana fruit pulp was closely related to the increase in the water-soluble pectin content and the decrease in the acid-soluble pectin content. Similarly, lower pectin content (GalA%) in alcohol-insoluble residues was reported in softened avocados (*Persea americana* Mill.). In the present study, the firmness of banana peel decreased significantly after softening. However, both biochemical and histological analyses revealed that the YM fruits even contained higher levels of pectin than GM fruits, though a significant increase in activities of PME, PG, and PL was observed. This might be due to strong activity in HG biosynthesis taking place in the soft YM fruits. Similarly, the GalA content in the cell wall of apple also increased concomitant with a decrease in cell wall stiffness after being subjected to ultrasounds [65]. These results likely suggested that some special types of pectin, instead of the total pectin content, were contributed to the firmness of fruit peel, while it might be varied with plant species and cultivars.

On the other hand, the pectin DM of YM fruit was lower than that of GM fruits, which might mainly result from the significant increase in PME activity. The removal of the methylesters from HG by PME during fruit development and ripening not only could facilitate PG action to increase cell wall porosity and decrease cell adhesion but also favor cell adhesion and the rigidity of the pectin network [66], and thereafter affect fruit texture. Except for the DM of pectin, the availability of Ca^2+^ also could influence the formation of cross-linked pectin gels, and they both contribute to tissue firmness. In the present study, the HG components recognized by JIM7, CCRC-M38, and LM18 antibodies showed significantly lower levels in YM fruits than in GM fruits, and the intensity of 2F4 antibody recognizing un-esterified/Calcium ion cross-linked HG also decreased 20%. As mentioned above, the levels of these antigens showed the same trends as the firmness of banana peel during fruit development. Thus, the lower DM, lower levels of Ca^2+^ cross-linked HG, as well as unesterified or deesterified to partially methylesterified HG were contributed to the reduction in firmness of YM fruits.

## 4. Materials and Methods

### 4.1. Plant Materials

The fruits of dwarf banana (*Musa* AAA, cv. Huanong Aijiao 1) at 0 d, 35 d, 60 d, and 85 d (harvested at the 85–90% plump stage, GM fruits) were collected from the field at the same time when the daily average temperature is approximately 25 °C. The fruit peel was isolated and frozen completely in liquid N_2_ before storage at −80 °C till use. Additional GM fruit samples were kept at 18–22 °C and 100% of humidity after treatment with 500 μL/L of ethylene (40% *v/w*) to initiate ripening. Twenty-four hours later, the humidity was lowered to 80–90% while the temperature was kept the same. The fruit became soft, and the peel became yellow 6 d after ethylene treatment, namely YM fruits. These fruits were also sampled for analysis. Each treatment consisted of three biological replicates, each containing six fruit fingers from independent plants.

### 4.2. Firmness of Banana Peel

Fruit peel firmness was assayed using a texture analyzer (TMS-PRO, Food Technology Corporation, https://www.foodtechcorp.com/ftc-tv/tms-pro-0, accessed on 28 August 2021) according to the manufacturer’s instructions. The compression force is measured at the maximum peak of the force recorded on the graph and expressed in *N*.

### 4.3. RNA-Seq Data Analysis

RNA preparation, library preparation for RNA-Seq, and data analyses were carried out as described previously [67]. In brief, we use hisat2 to map sequencing reads to the banana genome [68]. After that, featurecount software was employed to summarize the counts of each gene [69]. DESeq2 package was used for differential gene expression analysis [70]. We used a false discovery rate to determine the threshold of the *p*-value in multiple tests and analyses. In the present study, |log2 (fold change)| > 1 and a threshold of false discovery rate < 0.05 were used as the threshold to evaluate the significance of DEGs. GSEAPreranked was employed for gene set functional enrichment analysis [71]. Heatmaps are generated from TBtools software [72]. All RNA-Seq data are available from https://bigd.big.ac.cn/gsa/browse/CRA005631, accessed on 28 August 2021.

### 4.4. qPCR

Total RNA was extracted from the samples using the hot borate method described by Wan et al. [73], followed by convertomg to cDNA using HiScript III RT SuperMix for qPCR (+gDNA wiper) (Vazyme, Nanjing) according to the manufacturer’s guidelines. qPCR reactions were carried out as described by Chen et al. [74]. The relative expression levels of target genes were calculated with formula 2^−^^△△^^CT^ method. The primers used for qPCR are listed in Appendix A.

### 4.5. Activity Assay of HG Degradation Enzymes

The extraction of the crude enzyme was adopted from the method described by Mirshekari et al. [32] with slight modifications. In brief, fresh banana peel samples (1 g) were homogenized on ice with 2 mL of pre-cooled extraction buffer (0.1 mol/L sodium citrate, pH 4.6, buffer containing 1 mol/L NaCl, 13 mmol/L ethylene diamine tetraacetic acid, 10 mmol/L mercaptoethanol, and 1% (*w/v*) polyvinylpyrrolidone followed incubation at 4 °C for 1 h with occasional stirring. The extracts were centrifuged at 15000 rpm for 30 min at 4 °C and the supernatant was collected to assay the activities of PME, PG, and PL.

PME activity of the supernatant was determined using a modification of the method developed by Marcus and Schejter [75], exactly as described in our previous work [76]. The activity unit of PME was defined as the number of microequivalents/mmol of carboxyl groups cleaved by 1 mg of enzyme min^−1^, at 30 °C and pH 6.5. PG activity was determined according to the modified method of Pathak and Sanwal [77]: 100 μL of enzyme extract was mixed with 200 μL of 1% pectin (Sigma-aldrich, Chemical Co., St. Louis, MO, USA) in 50 mmol/L sodium acetate (pH 4.5) and 300 μL of 0.2 mol/L sodium acetate buffer (pH 4.5) followed by incubation at 37 °C for 1 h. To measure the amount of reducing sugar released, 1.5 mL of dinitrosalicylic acid was added to the reaction mixture [78]. After the reaction mixture was boiled for 5 min and cooled to room temperature, absorbance was measured at a wavelength of 540 nm (Infinite M200, Tecan, Switherland). PL activity was assayed by measuring the increase in reducing sugar released from polygalacturonic acid (Sigma-Aldrich, Chemical Co., St. Louis, MO, USA) [78]. The activity was analyzed in a reaction mixture containing 300 μL of 0.6% polygalacturonic acid (in 20 mmol/L sodium acetate, pH 4.5), 600 μL of 20 mmol/L sodium acetate (pH 4.5), and 100 μL of enzyme extract. After incubated at 37 °C for 30 min, 0.5 mL of dinitrosalicylic acid was added to the mixture before heated in boiling water for 5 min to stop the reaction. The absorbance was also measured at 540 nm. Specific activity was expressed as units. One unit was calculated as mmol/L of GalA catalyzed by 1 mg protein per hour. The protein content was determined by the method of Bradford [79].

### 4.6. Content of Pectin, Methanol, and DM

Measurement of pectin and methanol was based on the obtained alcohol-insoluble residue, which was prepared as described by Louvet et al. [80] with minor modification. In detail, approximately 2.0 g fresh samples were washed three times with 70% ethanol (*v*/*v*) at 70 °C for 30 min after homogenization. The pellet was crushed in liquid nitrogen and dried at 60 °C for 3 d. The determination of both pectin and methanol was carried out by the method described by Ma et al. [76]. The DM was calculated as the ratio of methanol content to 1 mole of GalA.

### 4.7. Immuno-Labeling of Antibodies Related to Pectin HG

The immuno-labeling of all antibodies was carried out exactly as described by Xu et al. [81], exceptfor the use of the 2F4 antibody. The labeling protocol of the 2F4 antibody was referred to in our previous work [82]. Sections (12 μm) were mounted on microscope slides coated with 0.2% polyethylenimine, dewaxed in absolute ethanol, rehydrated in ethanol-phosphate buffered saline (PBS) series, and washed in PBS. The primary antibodies used in the present study and the corresponding second antibodies are listed in Appendix A. Three biological replicates were prepared for each treatment. An Axio Imager D2 was used to examine the fluorescence, and ZEN software was employed to quantify the fluorescence.

### 4.8. Histological Study

RR (0.05% *w/v*) (Sigma-Aldrich, Chemical Co., St. Louis, MO, USA) in PBS buffer was used to stain de-methyl-esterified pectin. The sections were prepared, dewaxed in absolute ethanol, rehydrated in ethanol-PBS series, and washed in PBS for immuno-labeling fluorescence. RR was dropped onto the sections and kept at room temperature for 30 min. After rinsing in PBS 3 times, the sections were mounted in immersion oil (Ted Pella, USA) before observation with an Axio Imager D2 (ZEISS, Oberkochen, Germany). To stain all pectin, the sections were first treated with 0.1 mol/L NaOH for 3 min before RR staining.

### 4.9. Statistical Analysis and the Calculation of Correlation Coefficient

We perform statistical analyses using analysis of variance (ANOVA) in the statistical program SPSS 19.0 for Windows (SPSS Inc., Chicago, IL, USA). Three replicates were set for each treatment. Data are presented as the mean ± SE. Duncan’s multiple range tests are employed to evaluate multiple differences among means at a 5% probability level.

We use the “cor.test” function in R (version 4) to calculate the correlation coefficient between banana fruit peel firmness and enzyme activity.

## 5. Conclusions

Banana is one of the most important foods and fruit crops in the world. It is well-known plant cell walls play a very important role in fruit development and ripening. As a typical climacteric fruit, the disassembly of banana fruits during the ripening stage has been extensively studied. However, the associated information during banana fruit development and the organization of HG components in situ and their modification during the ripening process are unavailable.

In the present study, all differentially expressed *PMEs*, *proPMEs**, PGIs*, and *PLs*, most *GAUTs*, *PGs**, PMEIs,* and *PAEs* in banana peel showed a decline in expression during the fruit development, concomitant to the relatively stable activities of HG degradation enzymes and pectin DM. Four MaPGs (Ma09_g22850, Ma11_g02770, and Ma04_g08510 and Ma07_g16630) and three MaPAEs (Ma07_g07270, Ma07_g07240, and Ma07_g07250) might be involved in the expansion of fruit peel during banana development. All the investigated HG-modifying gene families showed differential expression patterns after ethylene treatment except for the PGI gene family. Six out of 10 top genes in the Core Enrichment Gene Set belong to HG degradation genes, and all of them were upregulated after softening, paralleled to the significant increase in activities of HG degradation enzymes and decline in fruit firmness. Interestingly, active HG biosynthesis took place during the fruit ripening process. The lower DM, lower levels of Ca^2+^ cross-linked HG, as well as un-esterified or de-esterified to partially methyesterified HG might be related to the reduction in fruit firmness after softening. The content of fully de-esterified HG recognized by CCRC-M38 antibody, instead of total pectin content, was positively correlated with the firmness of banana peel during both development and ripening processes. These results suggested that the HG-modifying genes are collaborated in the expansion and growth of cells in banana peels during fruit development and played a key role in the disassembly of pectin polysaccharides in fruit ripening.

## Figures and Tables

**Figure 1 ijms-23-00243-f001:**
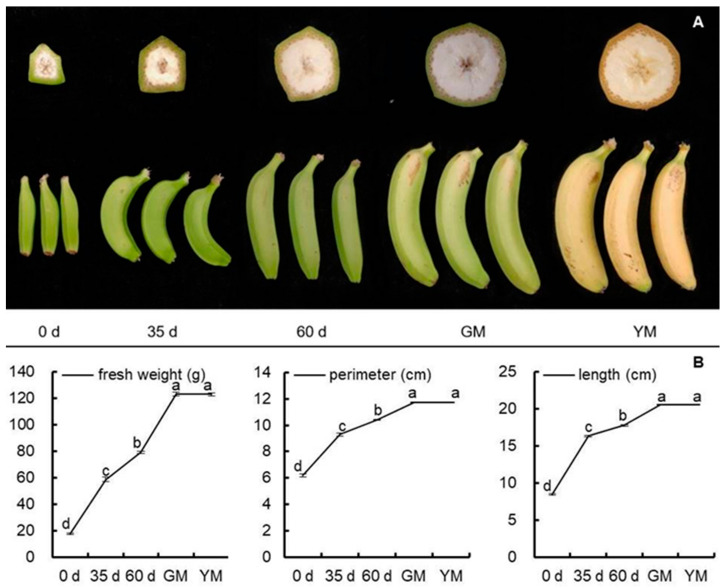
Changes in appearance and size of banana (*Musa* spp. AAA) fruits during development and ripening. (**A**) Fruit appearance and cross-section. (**B**) Fruit fresh weight, perimeter, and length. Each treatment consisted of three biological replicates. Data represent an average of three replicates ±SE. Values followed by the different letters represent significant differences using Duncan’s multiple range test at *p* < 0.05 after angular transformation of the data. 0 d: fruits just emerging from the bunch; 35 d: 35 day-old fruits; 60 d: 60 day-old fruits; GM: green matured fruits (at harvest, 85 day-old fruits); YM: yellow matured fruits (6 d after ethylene treatment).

**Figure 2 ijms-23-00243-f002:**
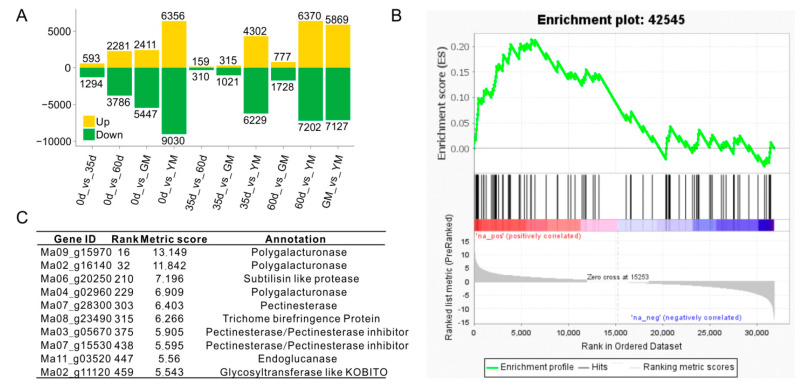
Functional analyses of differentially expressed genes in banana (*Musa* spp. AAA) peel during fruit development and ripening. (**A**) Numbers of genes upregulated and downregulated in comparison between every two stages; (**B**) Gene Set Enrichment Analysis result of cell wall modification; (**C**) Top 10 genes in Core Enrichment Gene Set, which includes 3874 members in banana genome. 0 d: fruits just emerging from the bunch; 35 d: 35 day-old fruits; 60 d: 60 day-old fruits; GM: green matured fruits (at harvest, 85 day-old fruits); YM: yellow matured fruits (6 d after ethylene treatment).

**Figure 3 ijms-23-00243-f003:**
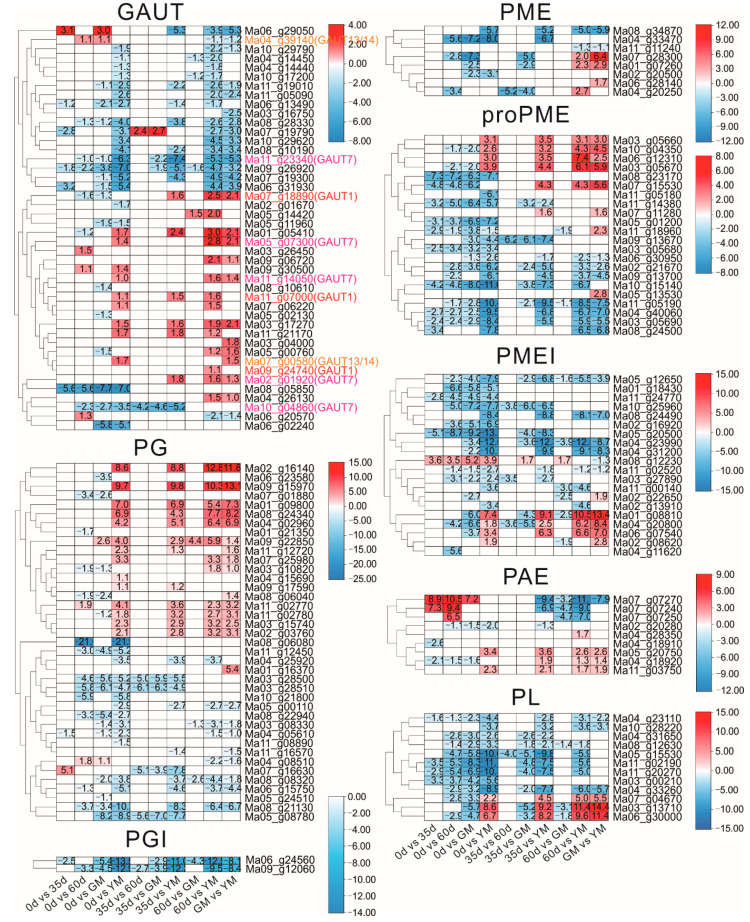
Differentially expressed homogalacturonan-modifying genes in banana (*Musa* spp. AAA) peel during fruit development and ripening. 0 d: fruits just emerging from the bunch; 35 d: 35 day-old fruits; 60 d: 60 day-old fruits; GM: green matured fruits (at harvest, 85 day-old fruits); YM: yellow matured fruits (6 d after ethylene treatment). GAUT: alpha-1,4-galacturonosyltransferase; PAE: pectin acetylesterases; PG: polygalacturonase; PGI: polygalacturonase inhibitor; PME: pectin methyl esterase; PMEI: pectin methyl esterase inhibitor; RNA-Seq: RNA sequencing. Heatmaps are generated from TBtools software with red and blue colors representing upregulated and downregulated, respectively. Among them, several GAUT genes, whose homologies in Arabidopsis play clear roles in cell wall modification, are colored in red, purple, and orange.

**Figure 4 ijms-23-00243-f004:**
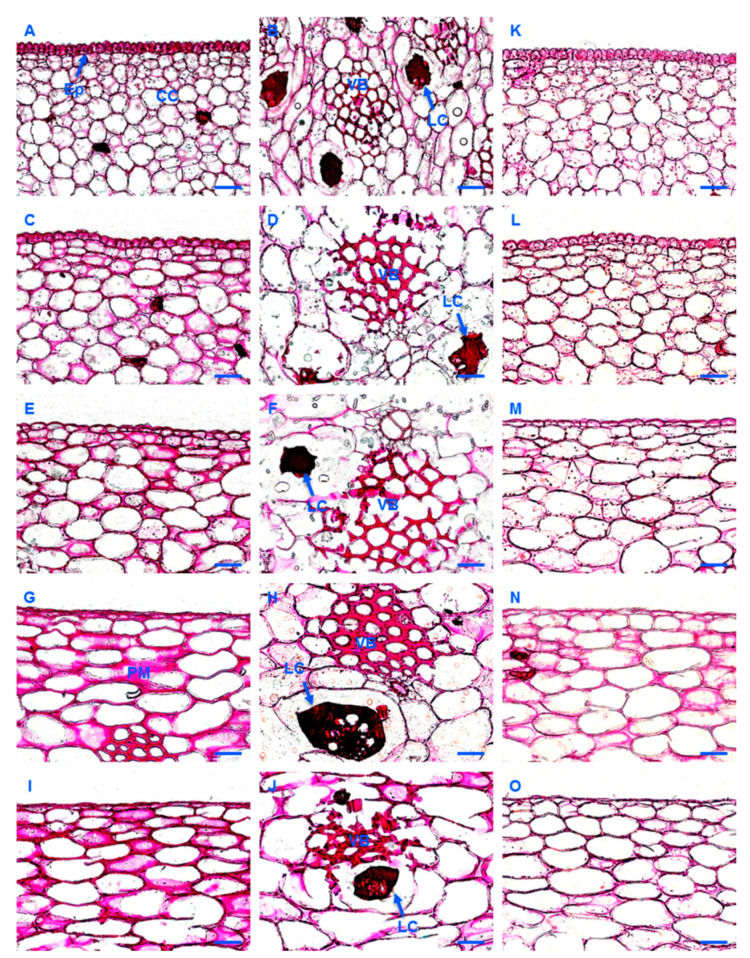
Histological analysis of pectin (**A**–**J**) and de-methyl-esterified pectin (**K**–**O**) in banana (*Musa* spp. AAA) peel during fruit development and ripening. (**A**–**J**) Sections stained with ruthenium red after NaOH treatment; (**K**–**O**) Sections stained with ruthenium red; (**A**,**B**,**K**) Peel of fruits just emerging from the bunch; (**C**,**D**,**L**) Peel of 35 day-old fruits; (**E**,**F**,**M**) Peel of 60 day-old fruits; (**G**,**H**,**N**) Fruit peel at harvest; (**I**,**J**,**O**) Fruit peel 6 d after ethylene treatment. Ep, epidermis; LC, latex cells; PM, plasma membrane; VB, vascular bundle. Bar = 50 μm.

**Figure 5 ijms-23-00243-f005:**
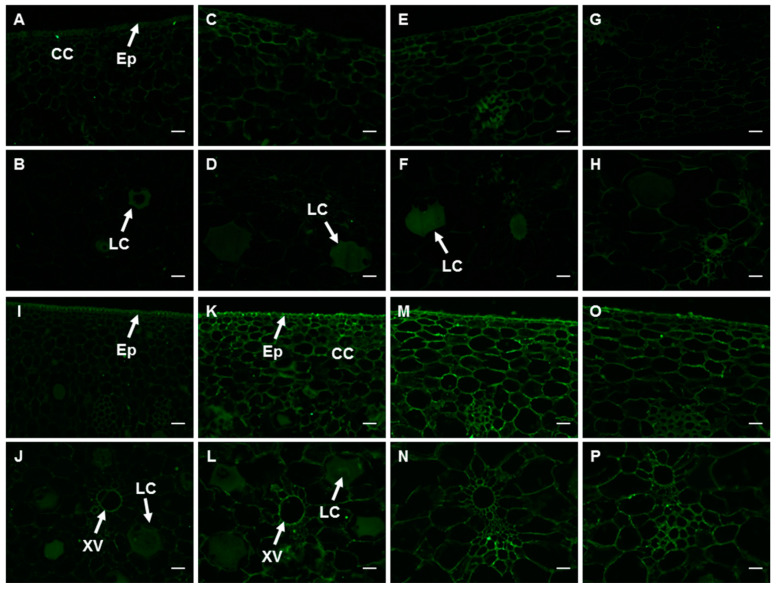
Spatio-temporal changes in PME (**A**–**H**) and the epitope of 2F4 antibody (**I**–**P**) in banana (*Musa* spp. AAA) peel during fruit development and ripening. (**A**,**B**,**I**,**J**) Peel of fruits just emerging from the bunch; (**C**,**D**,**K**,**L**) Fruit peel at 35 d; (**E**,**F**,**M**,**N**) Fruit peel at 60 d; (**G**,**H**,**O**,**P**) Fruit peel at harvest. CC: cortex cells; Ep: epidermis; LC: latex cells; XV: xylem vessel. Bar = 50 μm.

**Figure 6 ijms-23-00243-f006:**
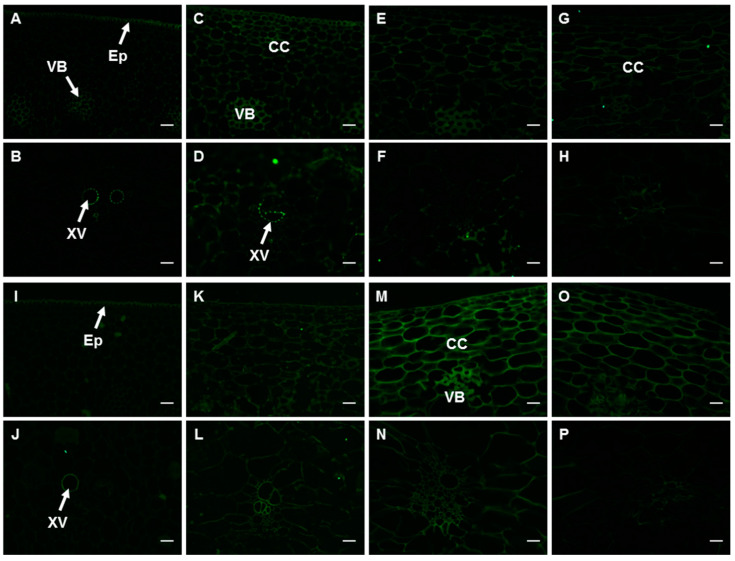
Spatio-temporal changes in the epitopes of JIM5 (**A**–**H**) and JIM7 antibodies (**I**–**P**) in banana (*Musa* spp. AAA) peel during fruit development and ripening. (**A**,**B**,**I**,**J**) Peel of fruits just emerging from the bunch; (**C**,**D**,**K**,**L**) Fruit peel at 35 d; (**E**,**F**,**M**,**N**) Fruit peel at harvest; (**G**,**H**,**O**,**P**) Fruit peel after ripening. CC: cortex cells; Ep: epidermis; XV: xylem vessel; VB: vascular bundle. Bar = 50 μm.

**Figure 7 ijms-23-00243-f007:**
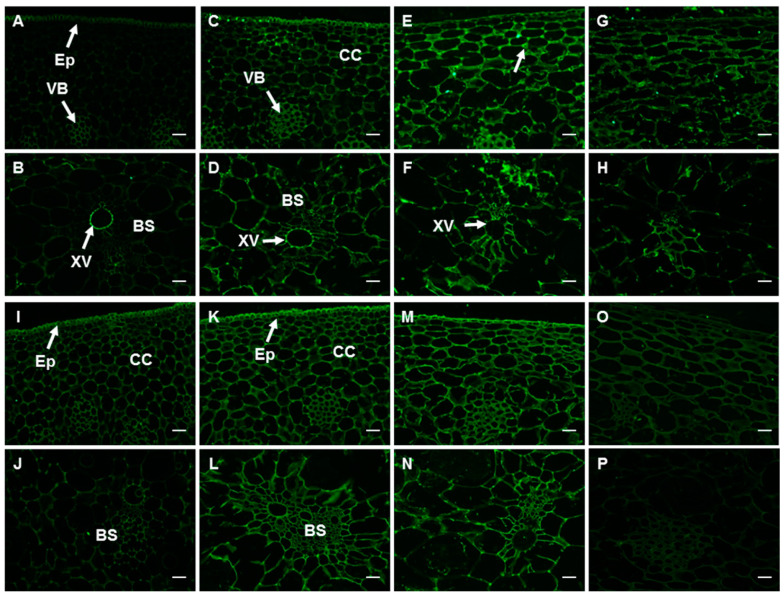
Spatio-temporal changes in the epitopes of LM18 (**A**–**H**) and CCRC-M38 antibodies (**I**–**P**) in banana (*Musa* spp. AAA) peel during fruit development and ripening.(**A**,**B**,**I**,**J**) Peel of fruits just emerging from the bunch; (**C**,**D**,**K**,**L**) Fruit peel at 35 d; (**E**,**F**,**M**,**N**) Fruit peel at harvest; (**G**,**H**,**O**,**P**) Fruit peel after ripening. CC: cortex cells; Ep: epidermis; LC: latex cells; XV: xylem vessel; BS: vascular bundle sheath; VB: vascular bundle. Bar = 50 μm.

**Table 1 ijms-23-00243-t001:** The changes in the activities of HG degradation enzymes and fruit properties in banana (*Musa* spp. AAA) peel during fruit development and ripening.

	0 d	35 d	60 d	GM	YM
PME activity (U)	15.59 ± 2.69 ^ab^	11.09 ± 1.72 ^b^	12.09 ± 1.78 ^b^	13.09 ± 1.30 ^b^	21.10 ± 2.95 ^a^
PL activity (U)	37.29 ± 1.61 ^bc^	28.78 ± 7.58 ^c^	38.31 ± 5.85 ^bc^	65.07 ± 10.29 ^b^	146.13 ± 15.79 ^a^
PG activity (U)	55.92 ± 12.92 ^b^	60.76 ± 5.36 ^b^	46.80 ± 15.64 ^b^	98.57 ± 24.28 ^b^	650.51 ± 121.35 ^a^
Pectin content mg/g(dry weight)	62.70 ± 0.22 ^a^	57.91 ± 2.82 ^ab^	53.40 ± 2.78 ^ab^	49.26 ± 2.36 ^b^	57.83 ± 2.41 ^ab^
DM (%)	59.07 ± 3.29 ^a^	48.98 ± 2.45 ^a^	51.34 ± 5.95 ^a^	52.85 ± 1.51 ^a^	47.60 ± 4.58 ^a^
Peel firmness (N)	10.90 ± 0.58 ^b^	12.20 ± 0.58 ^a^	12.70 ± 0.26 ^a^	10.77 ± 0.53 ^b^	4.93 ± 0.24^c^

Data represent an average of three replicates ±SE. Values followed by different letters represent significant differences using Duncan’s multiple range test at *p* < 0.05 after angular transformation of the data. 0 d: fruits just emerging from the bunch; 35 d: 35 day-old fruits; 60 d: 60 day-old fruits; GM: green matured fruits (at harvest, 85 day-old fruits); YM: yellow matured fruits (6 d after ethylene treatment). DM: the degree of pectin methylesterification; PG: polygalacturonase; PL: pectate lyase; PME: pectin methylesterase.

## Data Availability

The datasets supporting the conclusions of this article are included within the article (and its Appendix A).

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
