# Peer review of "Changes in Homogalacturonan Metabolism in Banana Peel during Fruit Development and Ripening"

_ijms, 2021, doi:10.3390/ijms23010243_

Round 1

Reviewer 1 Report

In sum, the manuscript was well organized and well written. In addition, the authors present a very interesting topic. There are currently many interests in the banana peel during fruit development and ripening. It would be of wide interest to the plant community, the fruit industry, and the international journal of molecular science readers.  However, I have some concerns about the manuscript, before publication:

The authors forgot to present the figure 2, which is very interesting and very important data for the manuscript. Therefore, it is very difficult to evaluate the current version of the manuscript, although the topic is very interesting. It is fair for the authors to provide the figure 2 and re-evaluate the manuscript for the publication in and the international journal of molecular science.

Reviewer 2 Report

The manuscript titled 'Changes in homogalacturonan metabolism in banana peel during fruit development and ripening' overall provides a good amount of data but some major concerns need to be addressed. The manuscript needs to be critically revised before acceptance. I have mentioned them below.  

Critical comments

  1. The abstract needs to rewrite again. The abstract should be a single paragraph
  2. The significance and novelty are missing in the abstract.
  3. Fig 2 is missing from the text. The remaining all figures need to be exhaustively re-arranged.
  4. The results need to be explained in a coherent manner. Some of the results are explained in the discussion section. This is not quite a right way.
  5. In results, the expression profile compares number of genes up/down-regulated, but there is no mention of specific genes between different stages, which is a very important part of the work.
  6. After specifying genes related to HG metabolism, it needs to validate them using qRT-PCR.

Minor Comments-

Introduction

Among them, PMEs are responsible for the subsequent de-esterification [25], PGs and PLs for further degradation of PMEs demethylated polygalacturonans [19].- Needs to rewrite this sentence.

Results

2.1

On what basis do you consider the zero-days; you could provide a reference and describe the length and size details/weight at different stages for a particular variety or certain range.

There are no scale bars provided to Figure 1 to evaluate different sizes and it has incomplete figure legends.

2.2

Expression profiles of each successive stage are similar as only 1,884 (593 up-regulated and 1,294 down-regulated), 469 (159 up-regulated and 310 down-regulated) and 2,505 DEGs (777 up-regulated and 1,728 down-regulated) were identified in comparison between 0 d and 35 d, 35 d and 60 d, 60 d and GM, respectively- What does mean similar here. Please clarify

2.4

YM fruits showed 6.60-13.90 times higher of PG activity than fruits at earlier developmental stages- In this sentence, the range of PG activity is very higher (6.60-13.90). Is it statistically significant?     

2.5

Figure 3.

J: Sections stained with ruthenium red after NaOH treatment;- Here, Do you mean A-J?

2.6.1

The antibody intensity decreased to the level of the fruits at 0d (Fig. S2).- Here, you mention that the intensity level decreased to the level of the fruits at 0d. But by looking at Fig 4 intensity looks more than day 0. Please verify intensity levels again.

2.6.2

Similar to PME, the JIM5 antibody intensity increased slightly with the development of fruits, but not at significant level (Fig. 5C-H, Fig. S2)- Here you said similar to PME, but in case of PME it not increased with development. Please check it again

The signal became stronger with the development of fruits and peaked at harvest, due to the stronger signal in the cortex cells adjacent to the epidermic cell and vascular bundles, showing antibody intensity of 21.36 which was more than two times higher than that of fruits at earlier developmental stages (Fig. 5K-P, Fig. S2).-  This sentence is not clear in its meaning. So rewrite again so one should understand clearly.

Discussion

Fruit Exocarpt- Do you mean Exocarp?

Round 2

Reviewer 1 Report

To sum up, the revised version was well organized and well written. Also, the authors present a very interesting topic. There are currently many interests in the banana peel during fruit development and ripening. It would be of wide interest to the plant community, the fruit industry, and the international journal of molecular science readers. However, I still have some concerns about the manuscript, before publication:
1. The manuscript was based on a large number of RNA-seq data, which is very interesting to the readers. To increase the readers' attention and the citation of the publication. The authors should upload the RNA-seq data to public databases, such as NCBI's database.
2. In Figure 1, the author should define the “fruit weight” as drought weight or fresh weight. Also, the figure legion in fig 1 is not well explained and the author should describe the Figures 1A and 1B. Furthermore, the author should describe the biological replicates in the figure 1B and the methods of statistical analysis, such as which p-value was used as the cutoff for significance, P<0.05 or P<0.01
3. Figure 2, authors should add more information about “Numbers of genes up-regulated and down-regulated in comparison between every two stages”. Please explain “between every two stages” in detail. About Figure 2C, the author should add the Gene Name in the form which is easy for readers to understand the data.
4. Please add the scale bar information in figure 4.
5. In the method section, 4.3 RNA-seq analysis. Please descript it in detail. The RNA-seq was analyzed using which software or R programming.
6. Please check the “Alignment” of Abbreviations at the end of the manuscript
7. The language of the manuscript needs to be improved

Reviewer 2 Report

The manuscript titled 'Changes in homogalacturonan metabolism in banana peel during fruit development and ripening' overall provides a good amount of data. The authors addressed most of the queries and provided satisfactory modification and corrections. Based on the authors answers to queries raised, and changes made in the article, the manuscript can be accepted for the publication.   
